# The psychological impact of torture and state repression in Türkiye between 2015 and 2018: Reports from Turkish refugees seeking asylum in Germany

**Estella Alejandra Tambini Stollwerck**[1]*, **Ilkem Sarikaya**[2], **Kathrin Yen**[2], **Hans-Christoph Friederich**[1], **Christoph Nikendei**[1]

1 Department for General Internal Medicine and Psychosomatics, Centre for Psychosocial Medicine, Heidelberg University Hospital, Heidelberg, Germany, 2 Institute of Forensic and Traffic Medicine, Heidelberg University Hospital, Heidelberg, Germany

* estella.tambinistollwerck@med.uni-heidelberg.de

**Data Availability Statement:** All the quotes that are cited in the main text are available in the

## Abstract

Torture seeks to undermine not only the physical and emotional well-being of an individual, but to damage the coherence of entire communities. Thus, torture and state repression are used to weaken entire subpopulations. After the failed coup d'état in Türkiye in 2016 and during the subsequently following state of emergency that lasted until 2018, allegations of torture and other degrading treatment in Türkiye spread widely. Since then, the number of asylum-seekers in Germany has risen considerably. This paper analyses the reports of twenty Turkish citizens that fled to Germany to seek asylum in the aftermath of the events. In semi-structured interviews held in Turkish, we assessed the experiences of torture and state repression, psychological consequences, and the current well-being and living situation. All interviewees described illicit violence of state authorities and government supporters, especially while under arrest. Though the methods varied, there was a constant pattern of imbalance of power. The psychological impact of these methods were present after relocation to Germany and included signs of PTSD, anxiety disorders, and major depression. The reports of torture, state repression, and their psychological impact emphasise the importance for policy makers to address the prevention of human rights violations and support the needs of survivors.

## Introduction

Torture is a dehumanizing crime that seeks to undermine an individual's personality [1]. The Convention against Torture and Other Cruel, Inhuman or Degrading Treatment or Punishment (UNCAT), Art. 1, defines torture as "any act by which severe pain or suffering, whether physical or mental, is intentionally inflicted on a person for such purposes as obtaining from him or a third person information or a confession, punishing him for an act he or a third person has committed (. . .) when such pain or suffering is inflicted by or at the instigation of (. . .)

original Turkish (or German) language in the supporting information file. The dataset is available in the Heidelberg Open Research Data (heiDATA) repository, permanent link: https://doi.org/10.11588/data/7O6QZO. It represents a minimal data set used to reach the conclusions drawn in the manuscript and provides the data required to replicate the reported study findings in their entirety.

**Funding:** The authors received no specific funding for this work.

**Competing interests:** The authors have declared that no competing interests exist.

a public official" [2]. State repression is understood as the use of fear tactics and the restriction of civil liberties of political opponents by government authorities, the political party in power, and its supporters [3]. As of June 2023, the UNCAT has 173 state parties [2]. Nevertheless, torture is still practiced in many countries today, having major social and psychological consequences for their societies. Among others, trauma sequelae disorders, such as post traumatic distress, major depression or anxiety disorders, are prevalent in persons that were exposed to state repression and torture [1, 4–6]. Survivors often not only report emotional symptoms, but also sleep disturbances, chronic pain, headaches, and difficulty in concentrating [5, 7]. Accordingly, in a comparison between tortured and non-tortured activists, torture raised negative psychological effects independent of other stressors in a politically repressive environment [8].

## Torture and torture-related practices in Türkiye

Türkiye is one of the European countries that support UNCAT but have allegedly practiced torture in the last decade [9–12]. Torture and other ill-treatment by the Turkish state towards political dissidents has been documented by the UN Subcommittee on the Prevention of Torture in 2017 and the Council of Europe Anti-Torture Committee [9, 10, 12, 13]. Meanwhile, the institutionalised use of repression and torture, especially against Kurds and left-wing citizens, was carried out long before [4, 14–16]. Allegations of gendered political violence, as well as rape and torture by police officers in custody in Türkiye are common [17, 18]. Amnesty International regularly denounces human rights violations and torture practices [19]. The Human Rights Foundation of Turkey (TİHV) describes in its 2022 annual report that torture has been used ubiquitously and systematically in Turkey for years, reporting 1201 survivors of torture approached their centres in 2022 [20]. Türkiye has signed the United Nations' Convention against Torture and the European Convention for the Prevention of Torture and is therefore obliged to adhere to the absolute ban on torture and to create bodies that take preventive action against possible torture practices. In its official responses to reports over the visits undertaken by the United Nations' Subcommittee on Prevention of Torture and Other Cruel, Inhuman or Degrading Treatment or Punishment (SPT) and the Committee against Torture (CAT), the Turkish government stresses the adherence to the international human rights system and the determination to take effective legislative, administrative and judicial measure for prevention, including the implementation of National Preventive Mechanisms [21, 22]. The Turkish government emphasises a 'zero tolerance policy against torture' which had been 'continuously and decisively implemented well over a decade, preventing all forms of torture and other cruel, inhuman or degrading treatment or punishment' [21; p. 11].

State repression against ethnic minorities, particularly its large Kurdish population, has led to widespread claims of human rights abuses and a large number of political refugees claiming to have suffered torture [4, 18]. Furthermore, the criminalisation of the petition signed by the so called "Academics of Peace" was seen as a massive suppression of academic freedom [23–25]. In 2015, peace talks with the Kurdish minority ended and subsequently led to the deterioration of the relationship with the government. The overall security situation worsened as Türkiye was shaken by various suicide attacks in the following years claimed by the Daesh, among them a detonation of explosives at the "Labor, Peace and Democracy Rally" that took place on 10 October 2015 in Ankara [26]. At this occasion alone, more than 100 civilians were killed and over 500 people were injured. On 15 July 2016, an unsuccessful coup attempt by the military led to the imprisonment of intellectuals, purges in the Turkish military, and mass dismissals in the public sector [23, 25, 27, 28]. Subsequently, the government imposed a state of emergency that lasted for two years in which fundamental rights were restricted or even abolished altogether [23]. Entailed by these developments, Türkiye is one of ten nationalities with

the highest rate of asylum applications in Germany since 2017. From 2017 until 2019 alone, Germany registered around 29.000 applications for asylum from Turkish citizens, making it the third biggest group of new asylum seekers in 2019 [29].

## Literature review and the Istanbul protocol

Several studies examined the human rights violations in Türkiye in the aftermath of the coup attempt, focussing on education rights [30], loss of jobs and the freedom to travel [31], torture and the right to a fair trial [23], the consequences for the Hizmet (Gülen) movement [32], and discrimination and social isolation [33]. Aydin and Avincan's study collected several reports on torture, illegal interrogation techniques, naked body searches, and sexual harassment against all genders, as well as ill-treatment of family members. In a study with Kurdish torture survivors, Dehghan and Osella [34] examined the psychological impact of sexual torture and revealed the negative consequences for all forms of social relationships and the bodily self. In line with this research, Patel [35] published an urgent call for more efforts to assess the psychological impact of torture practices on survivors and their families.

For a psycho-medical assessment, the Istanbul Protocol on the Effective Investigation and Documentation of Torture and Other Cruel, Inhuman or Degrading Treatment or Punishment provides the most important tool, updated by four civil society organisations and UN anti-torture bodies [1, 36]. It offers guidelines for the effective investigation of torture and other inhuman treatment, including general considerations for interviews, the documentation of physical evidence and a psychological evaluation. The Istanbul Protocol consists of the following components: History of ill-treatment, current psychological complaints, post-torture and pre-torture history, as well as medical, psychiatric and substance use history, and the assessment of mental status and social function. Over the past 25 years, it has contributed to medical and legal investigations worldwide [36].

The aim of this qualitative research was to examine the psychological impact on vulnerable groups of Turkish citizens that have fled the country in the aftermath of the end of peace talks between the Turkish government and the Kurds, the failed coup d'état, and the prolonged state of emergency. Therefore, the presented study examines the following research questions:

a.  What reports do asylum-seekers in Germany give about their personal experiences of torture and state terror in Türkiye between 2015–2018?

b.  What was the mental health burden that arose due to these experiences?

## Methods

### Study design and ethical considerations

The study took place on the premises of the Heidelberg Institute of Forensic and Traffic Medicine in the first year of the COVID-19 pandemic. Our research team consisted of four medical doctors, two of them specialised in forensic medicine, the other two specialised in psychosomatics, and one clinical psychologist. Given the exploratory character of our research questions, we conducted a qualitative study with semi-structured interviews with Turkish asylum seekers and refugees on their experiences in Türkiye between 2015–2018 and the subsequent impact on previous and current mental health.

This study followed the ethical standards laid down in the 1964 Declaration of Helsinki about confidentiality, informed consent, and a guided research protocol. The interview questions are directly based on the recommendations of the Istanbul Protocol [1] for the

documentation of torture and other ill-treatment. Safety and confidentiality were established by conducting private interviews in a separated room. The study was approved by the Ethics Commission of the Medical Faculty at Heidelberg University (S170/2020; 30.03.2020). All participants were fully informed about the aims and methods of the study and completed an informed consent form prior to participation. There was no reimbursement for participation, but travel expenses were covered.

## Participants and procedure

Our targeted group were asylum-seekers from Türkiye that fulfilled the following inclusion criteria: Age of 18 or older and residence in Türkiye for at least one year during the period of interest from 24 July 2015 until 18 July 2018. We searched for participants with messages on notice boards, via non-governmental organisations, and through personal contact to the local Turkish community from 02 May 2020 until 15 October 2020. The researchers handed out information and the Harvard Trauma Questionnaire (HTQ) upon request [37]. Eligible participants filled out the HTQ to assess an indication of experiences of torture and state repression between July 2015 until the official end of the state of emergency in July 2018. If the individual was willing to participate in the study and fulfilled all requirements, they contacted the second author (I.S.) and arranged a meeting on the premises of the Institute for Forensic and Traffic Medicine of the University of Heidelberg. We included all Turkish refugees that answered to our posting and expressed to have experienced torture in their home country.

Upon arrival, all participants signed a form that stated they had no symptoms of Covid-19 and the medical researcher (I.S.) measured their temperature. Thereafter, participants answered questions concerning sociodemographic data, such as age, nationality, religion, ethnicity, and educational level. All interviews were audio recorded and conducted in Turkish by the first authors (E.T. and I.S.), facilitated by German–Turkish interpretation in both directions by the second author (I.S.), who also clarified cultural queries if necessary. The Appendix offers the original version of all quotes.

## Data analysis

To obtain descriptive statistics, the analysis of demographic variables and baseline characteristics was managed with Excel, program version 16.75.2 [38]. All interviews had a length of one to two hours, including the time needed for interpretation Turkish—German. They were transcribed verbatim in Turkish following predefined transcription rules and translated into German by the second author (I.S.). We used the written translation of the original Turkish text parts instead of the consecutive translation into German on the recording to stay as close as possible to the original statements. The qualitative data was analysed with MAXQDA 2022, Release 22.2.0 [39], led by the principles of qualitative content analysis following Mayring [40]. The first author developed a coding framework combining inductive and deductive themes, guided by a review of relevant literature on torture. Inductive codes were derived directly from the narratives. Each statement, either a single sentence or content related sentences, was assigned to our key research topics as a content analytic unit of analysis. Some phrases referred to more than one category, so double categorisation was possible. Each single or multiple content-bearing sentence was seen as a basic unit and was coded by a term that formed a relevant category. Afterwards, the categories were summarised into main themes until a final number of relevant main themes for all participants was found (commonly termed content saturation). If necessary, the main themes and categories were adjusted to form a common category [40].

## Results

A total of $N = 25$ participants filled out the HTQ, of which five participants declined participation due to safety concerns and fear of political consequences. $N = 20$ participants took part in the study. All but one participant, whose interview was conducted in German, provided their answers in Turkish. Table 1 shows the sample characteristics of all study participants. Most interviewees arrived in Germany between 2017–2019 and several were granted asylum for three years, but some were still waiting for a decision by the authorities.

Table 2 details the Key Themes and Codes that we derived from the interviews. We identified 1308 statements that were coded and summarised into categories. Finally, 14 categories within three main themes were derived and included in this publication.

### Experience of violence

The interviewees described their experiences with different forms of state repression and torture by the use of force, threats, and sexual assault. Table 3 indicates the frequencies for different forms of violence that participants marked in a checklist.

*Torture* (51 quotes): All interviewees described some form of torture, either psychologically or physically. Some witnessed physical violence towards friends and family in prison or in public spaces, others experienced it themselves. Eleven out of twenty interviewees described torture in detention or after abduction with tremendous human rights violations. Some provided detailed depictions of torture with highly specialised means and knowledge used to obtain confessions or to stop people from being politically active. Some mentioned isolated cells that had been equipped with sound insulation, light that shone constantly, and meals being served at randomly varying times to affect the sense of time. Two participants reported that, occasionally, the guards played loud military music all day long and forced them to stay awake. One participant described being kept in an underground room for hours, where the ground emitted electric shocks and burned his feet. He experienced induced choking when a police officer

Table 1. Participant demographic information, $N = 20$.

| Gender | Percentage |
|---|---|
| female | 60% |
| male | 40% |
| diverse | 0% |
| **Age** | |
| 20–24 | 5% |
| 25–29 | 25% |
| 30–34 | 25% |
| 35–39 | 25% |
| 40–44 | 10% |
| 45–50 | 10% |
| **Religion and Ethnic Group** | |
| Alevis | 35% |
| Atheism | 10% |
| Islam | 20% |
| Other | 35% |
| Kurdish | 70% |

Note. The order is based on the alphabetical or numerical order except for the category of Gender.

**Table 2. Key themes and codes.**

| Main themes | Codes |
|---|---|
| **Experience of violence (8 categories)** | Torture<br>Arbitrariness and state violence<br>Police house searches<br>Violence at demonstrations<br>Detention and police custody, charges, and prison conditions<br>Violence in the context of conflicts in Kurdish areas<br>Violence of civil government supporters<br>Sexualised violence |
| **Psychological consequences (3)** | Acute response<br>Emotional reaction: Fear and anxiety; Anger and impulsivity; Powerlessness and hopelessness; Courage and strength<br>Long-term consequences: Hypervigilance and lack of trust; Isolation, loneliness, and silence; Suppressing feelings and forgetting |
| **Current Situation (3)** | Political conviction and religious belief<br>Subjective well-being, work, and social activities<br>Plans for the future |

knelt on his chest and larynx while pressing a wet towel on his nose. Some described the constant fear of being followed or under surveillance in a dangerous political environment as torture. The bombing at a peace rally in Ankara in 2015 was considered as torture, as several participants suspected that the state authority knew about the attacks without intervening.

**Table 3. Types of violence reported by survivors of torture and state repression, N = 20.**

| Type of violence | Number of participants |
|---|---|
| 1. Solitary confinement | 13 |
| 2. Sleep deprivation | 15 |
| 3. Stress positions | 14 |
| 4. Electric shocks | 4 |
| 5. Falaqa | 2 |
| 6. Palastenian Hanging | 0 |
| 7. Prolonged wearing of a blindfold or handcuffs | 15 |
| 8. Extreme hunger or thirst | 12 |
| 9. Exposure to extreme brightness or total darkness | 15 |
| 10. Exposure to freezing water or forced submersion | 4 |
| 11. Forced injection of drugs | 1 |
| 12. Threats with dangerous animals | 0 |
| 13. Mock executions | 7 |
| 14. Persecution perceived as life-threatening | 17 |
| 15. Repeated threats of violence | 16 |
| 16. Naked and invasive body searches | 16 |
| 17. Sexual assaults and threats<br> a. Touched on genitals<br> b. Forced to undress<br> c. 'Virginity checks'<br> d. Rape with objects | <br>11<br>14<br>1<br>2 |
| 18. Were fellow inmates forced to commit violent acts? | 6 |
| 19. Were you forced to commit acts of violence against others? | 1 |
| 20. Did you feel dehumanised or degraded by the state? | 16 |

*As I said, one of the methods of torture is, for example, the national anthem (. . .). But there they play it, play it again, play it again, play it again, play an anthem, then the army march [Mehter Marşı], and you go crazy. And that loud noise, they are trying to drive you crazy, they are brainwashing you. (TN 21, Pos. 35)*

*For 93 days, what I experienced and heard, the questions I was asked, the attitudes of the people who asked me questions, their professional experience, the technological possibilities they had access to, I was taken to a torture centre where each of the scientific torture methods were applied, not a gang, not three or five punks, but an organised torture centre with changing shifts. I already had in my mind "maybe I will be killed" . . . (TN 21, Pos. 36)*

*Arbitrariness and state violence* (57 quotes): Interviewees described various forms of police assaults. Nearly all participants described physical and psychological violence as threats, insults, or punches when they were forced to provide fingerprints in state prisons. As a formal requirement, interviewees described a medical examination before and after their custody. Although this procedure was formally followed in most cases, it often resulted in the injuries at hand being intentionally ignored. Only one participant stated that the doctor resisted the police and declined to write a report. In public spaces or online, some stated they were threatened and told to stop their political activism. Some learned that their telephone conversations had been tapped for months to years. Two interviewees reported they had been kidnapped by special forces and were interrogated by them. Several participants said specifically that, in order to cause fear, police officials described the violence they were about to inflict upon them. Others felt that the level of repression rose after the failed coup attempt.

*The police entered the university; we did some small political work at that time . . . They threatened me, saying "your father and brother are imprisoned, we will arrest you, too" . . . (TN 12, Pos. 19)*

*They beat you and frightened you there too, but not as much as the second time, the second time, in October, when there was an attack on Rojova, Türkiye's attack on Rojova, they said "we will take you to the border, we will shoot you in the head, we will say that we killed you because you are a terrorist". They said they would do that now. (TN 16, Pos. 61)*

*Police house searches* (15 quotes): Many interviewees described police house searches as unjustified and as a serious invasion of privacy. Some said that police officials had entered the house by smashing the door, put the residents on the floor, and stomping them with their feet. For several participants, this dominant behaviour caused high levels of psychological stress and a feeling of constant insecurity. Some interviewees stated that police officers threatened family members and caused an insecure environment, which they felt had a high impact on neighbours who were intimidated and afraid to approach the family afterwards. One threw away copies of books and newspapers in the night of the coup attempt and thereafter in fear of oppression.

*At first, we were subjected to the screams of our friends in the next room as a result of the tortures inflicted on them. Psychological torture such as "We will come to you soon, we will sweat on you, then we will make you scream the same screams". So, in addition to the beatings, punches, kicks and hair pulling. Then psychological torture such as showing us the bloodied bodies of our friends in the next room and saying "take a good look at this, you will never see it again" and making us fear that he would be killed. (TN 2, Pos. 45)*

In the incident in Istanbul, for example, after the coup, our house was raided many times, our doors were broken down in the morning, inside, I mean, very arbitrary treatment, they scattered everything, the books I read were officially printed books, books that were not forbidden but were forbidden according to them. (TN 5, Pos. 5)

*They took turns passing by my mum, swearing at her. My mum still doesn't say the swear words. She just says "they told me very bad things about you". I don't know what they said. Then they got in line and spat in my mum's face. (TN 10, Pos. 93–94)*

*Violence at demonstrations* (28 quotes) and at the peace rally in Ankara 2015 (22 quotes): Interviewees described several occasions when they experienced oppression by the state police at peaceful demonstrations, others observed violence between protestors and state officials. The allegations of various participants in the context of the peace rally in Ankara 2015 concerned delays and failure to provide aid in the aftermath. Several interviewees accused the secret service of knowing about the suicide bombings without publishing a warning to the population, pointing out that the police were absent during the beginning of the protest march, which was unusual for such a big event. After the bombing, various interviewees depicted horrific scenes of blood all across the square and human remains everywhere on the ground, even on the clothes and hair of survivors.

*We already call the bombing there a terrorist state. Apart from that, I think people are being tortured psychologically, because if people are still psychologically unable to come to their senses, if they commit suicide, it means that both forms of torture exist. (TN 17, Pos. 75–76)*

*There was always the danger, getting caught . . . For example . . . If I were in Türkiye now, I would be caught, I would be put in jail, but no information would be presented anywhere. I mean, the police would say, "this man has done this, he has done that". Therefore, there is already such distrust and psychological pressure on everyone. This is currently being applied. The man can lock up whoever he wants, kill whoever he wants, but there are also these intense things I mentioned. I grew up with these stories. You can listen to thousands of these stories. (TN 15, Pos. 26)*

*Detention and police custody, charges, and prison conditions* (126 quotes): The majority mentioned that the police had taken them into temporary custody. Indeed, the interviewees described that so many people were taken to custody that it started to feel almost ordinary to them. In custody, they mentioned different lengths of stay under custody, from several hours to weeks, during which the prisons offered little space, food or water. Sometimes they weren't allowed to use the toilet or shower. Some interviewees also felt that wardens deliberately spread distrust among detainees, used body searches to humiliate them or isolated them completely. Another participant described complete darkness in isolation and that calls and visits were prohibited. Under police custody, several interviewees reported that they were not allowed to contact family and friends, spent hours in painful handcuffs, and that constant intentional disturbances led to no rest. Some felt that soldiers and police officers tried to humiliate them to make them feel small and to prove their authority. One described that they were forced to slap each other in the face. One participant mentioned that she was in an isolation cell for one year. Another described that there were no beds and people had to sleep on cardboard boxes on the floor in overcrowded rooms.

*We didn't want to give our fingerprints and were discussing it . . . They said, "Either you give your prints voluntarily or we will take them by breaking your fingers." (TN 1, Pos. 31–34)*

*Similarly, you go to the forensic medicine, the doctor there also plays his part in this, even if not directly physically. I mean, you've got bruises all over you, there is blood coming out of your nose, I don't know, some people's eyes are closed, and they can't be opened, but they can give you a healthy report without examining you at all. (TN 7, Pos. 23)*

*Violence in the context of conflicts in Kurdish areas* (38 quotes): Several interviewees described violence in the context of conflicts in the Kurdish areas, especially in violent conflicts and curfews in Cizre, but also in Nusaybin and Diyarbakir. Participants mentioned bombings and that civilians were burnt alive. Two interviewees recalled the public campaign and threats against the "Academics for Freedom", followed by their suspension, as a reaction to their petition to cease state violence in mainly Kurdish populated areas.

*Because what happened was not normal. The bodies were all burnt to ashes, like burning wood. The bodies were unrecognisable now, some of them still could not identify their bodies. Torture, okay, war is war, but war is on the mountain, not in the city. You kill the ones on the mountain, but so much torture, so much burning, it was all human; I don't know, you can't place it anywhere. (TN 17, Pos. 50)*

*Violence of civil government supporters* (25 quotes): Participants described right-wing government supporters set fire to Kurdish stores and entered private homes and pelted them with stones, sometimes with the support of local police. They stated they were at the mercy of constant harassment and persecution. One teacher reported that she was accused by families of her students of spreading Kurdish propaganda in her classes and was therefore threatened via social media.

*I received a message on Facebook saying, "We will wipe the Kurdish race off the face of the earth". A man I don't know personally sent me his genitals, he posted it as a picture, and it was a message like "I will make you moan under me until you say ' How lucky anyone can be who calls himself a Turk . (TN 13, Pos. 36)*

*I am convinced that the physical and psychological pressure outside is also torture, so there is definitely a connection between them. Because that is the perception and the mentality. There is a perception that "you can do whatever you want to those who are not of your kind, you can say whatever you want" and they believe that they are powerful. Of course, they are really powerful because they have the state behind them. That's why it's torture for me. (TN 13, Pos. 51)*

*Sexualised violence* (13 quotes): Participants, both female and male, described different forms of sexualised violence. Various participants described that there was constant harassment by police officers in and outside of prison and that they were touched inappropriately. One described specific insults because of her bisexuality, another received explicit pictures of male genitals and threats about being raped. As one of many torture methods, one male participant depicted he had to kneel with police officers positioned behind him, threatening to insert an object into him.

*There was sexual harassment at every detention. (TN 10, Pos. 60)*

I was afraid that they would rape me, that they would catch me somewhere, that they would do something to me . . . Because Türkiye is very convenient in this sense. You live alone, you are a single woman, you have been defamed by a very dangerous newspaper, and we know how dangerous their supporters are. (TN 13, Pos. 36)

*They wanted to force me to undress for the body search . . . They didn't manage to do that . . . But this was a threat, in between they constantly insulted me, like . . . about my sexual identity, for example. . ."Let the female police come if you want" and similar disgusting "jokes". (TN 1, Pos. 35)*

## Psychological consequences

Considering that all participants had experienced some form of state repression in Türkiye, every interviewee described their acute response in a situation of imminent danger, emotional reactions during and after the events, as well as general long-term consequences.

*Acute response* (51 quotes): Most interviewees mentioned some form of emotional or mental response to imminent danger. Many described being primarily afraid for relatives and friends who were in a situation of danger with them, but also for those from whom they were separated. Some reported that they tried to stay calm, take the perpetrator's perspective or simply find a way to save themselves and their acquaintances. Many interviewees felt responsible and guilty that their friends were in a difficult situation because of them. One participant stated that he suffered such agony due to physical violence in prison that he wished for them to kill him right away. Other participants reported that they did not even believe in survival. Some feared that officials would make their corpses disappear so that their relatives would never know of their death. Others described they only wished they would see their families again. Some participants stated they resisted, others deliberately did not take a defensive position, but pretended to be ignorant of the accusations. One described compliance to unnecessary police requirements out of fear.

*At that moment, I thought they should kill me right away so that I could get out of there . . . Psychologically, I was convinced that I would never get out of there. He also puts pressure on you by saying "you're here with us for another 14 days " . . . "You won't get out of here yet" . . . At that moment you think to yourself, "I might as well get killed now" (TN 4, Pos. 95–96)*

*In fact, you're not really thinking at that moment. You're just scared. There is fear, yes . . . There's no need to pretend you were not afraid just to seem brave. You think that everything is over. (. . .) So in that moment you are scared, in that moment you don't think anything, you don't feel anything. Maybe in a little moment of emptiness, you might think a little bit, because you don't have any chance to think . . . What will you even think about? (TN 10, Pos. 110–111)*

*Emotional reaction* (in total 236 quotes): Several participants reported emotional difficulties in association with the torture experienced. Some said they stopped feeling at all, some developed strength to handle the dangers they were confronted with. The most common themes mentioned were fear, anger, inner conflicts, powerlessness and hopelessness, sadness and grief, as well as shame and guilt. Many participants described grief for their losses and overwhelming sadness since they arrived in Germany. Some participants expressed sadness that they could not witness important life events, such as weddings or funerals, because they were in Germany. They pondered with their decision to leave Türkiye because they felt they had abandoned their family and friends. One participant stressed that he felt guilty that his children had to endure difficulties because of his decisions.

*Everything was meaningless to me now. I couldn't cry, I couldn't laugh, there was such an intermediate tone in my emotions (. . .) Everything was meaningless now. I started thinking,*

*"After this hour, even if all of Türkiye is liberated, even if this despotism is overthrown, there is no point anymore." Because terrible things had happened. There had been war crimes. (TN 15, Pos. 22)*

*I feel guilty towards them. Yes, anyone can experience this stress and troubles in our latitudes, but I was fighting for something, for the values I believe in. But I say, "I wish I hadn't included them". I wonder if there was ever a chance to not include them. (TN 16, Pos. 165)*

*Fear and anxiety* (46 quotes): Almost every participant reported to have experienced some form of fear because of state repression and torture. Some knew the prison that they were taken to from stories and were afraid of what would await them there. Once inside, they were afraid to be called into interrogation, one interviewee said. One participant described hiding his fear so as not to further frighten other (female) prison inmates. Fear was present in dangerous situations as when the state of emergency was proclaimed and in everyday life when threats reached them. Several interviewees highlighted that, eventually, it was the most prominent motivation for the flight to Germany. Some described inner unrest when they spot police officers even today. Others experienced panic attacks and were overwhelmed by fear when something reminded them of their frightful experiences.

*And we were all afraid that. . . Little did we know that it would be just those two suicide bombings. I was sure I was going to die that day. I was sure of it. (TN 11, pos. 26)*

*I was always anxious, to be honest. Unfortunately, there was a period when I had to constantly look to the right and to the left because something would happen at any moment, something would come out of somewhere. (TN13, Pos.43)*

*Anger and Impulsivity* (33 quotes): Many participants described that they exhibited higher levels of impulsivity and aggression after the events. Some hoped for revenge and demanded accountability, for several interviewees anger was a driver of motivation and hope. Others described that the injustices they had observed created a need to shout and express anger. Some said they did not express the anger, however, or directed it against themselves.

*For a while I was very angry, then that anger turned into sentimentality, then for a while I was unable to feel anything towards anyone or anything, including my parents. Now, I don't know if it's because of those periods, but for example, I'm a person who gets angry very quickly, who has an immediate outburst of anger, but I'm also extremely emotional. (TN 6, Pos. 101)*

*I mean, I had an extreme anger, and after a while that anger scabbed over. It wasn't a very painful anger anymore; it was something else. Nothing made sense anymore. (TN 15, Pos. 22)*

*Powerlessness and Hopelessness* (24 quotes): Some interviewees described that they were not able to process what was happening around them in dangerous situations, so they froze and could not react. Several participants experienced major hopelessness, or they felt despair like they were at a dead end. Some accused the European Union of not intervening.

*So, we waited, then we looked, there were some friends on the ground, some of them were injured, some of them were dead, these two friends . . . We didn't do anything, to be honest . . . I mean, I couldn't react . . . Because I didn't fully understand what was happening . . . (TN 3, Pos. 97)*

I remember during that time, from August onwards, when in Sur and Diyarbakir, in Ahmet, many people were . . . were killed. I remember that, I am not exaggerating, I had to cry every night. Always in my bed. I cried, cried, cried, I felt powerless. I thought to myself, it's not possible, I can't go on like this under these conditions. (TN 11, pos. 24)

*Now, people generally have good intentions and most of the society in Europe understands us. Not all people think evil, but the state is a different organism and all the states in Europe know very well what is going on, but they remain silent. They continue to trade with Türkiye, they continue to sell arms, they continue to support Erdogan financially. Because Europeans think very analytically, the states. So, if Türkiye experiences a serious chaos, refugees will start coming in a different way. Türkiye is currently a member of NATO. If it withdraws from it and turns towards Russia, the calculations of Europe and America will be mixed up. So, analytical in that sense. (TN 15, Pos. 24)*

*Courage and strength* (67 quotes): Numerous participants expressed that they had shown strength in moments of great danger. They had resisted provocations of police officers and kept calm to avoid greater difficulties. Some stated that their will to survive grew when they hit rock bottom and suffered intensely. Many interviewees stated they told themselves to be strong and brave because others would be worse off. One participant vigorously sought international courts to work for a just verdict regarding the events in Türkiye. His way to maintain strength, he said, was to always seek purpose. Many described that they wanted to stay in place to resist, but in the end it had not been possible, so they fled to Germany.

*And this loud noise, they are trying to drive you mad, they are brainwashing you. At that time, as one of the psychological resistances, I created a resistance by directing my mind to other things that I could occupy myself with, creating a little world in my head, in my mind, in that noise. (TN 21, Pos. 35)*

But after a while, the fear is gone . . . Courage replaces fear . . . I guess that's what you think . . . "I have to take precautions myself . . . What else can he do . . . The most he can do is kill . . . He's doing what he's going to do, what else is left?" (TN 4, Pos. 96)

When I saw that I was still alive after the fall, I realised that they hadn't thrown me out of the window. Do people still want to live after all they have been through? I did. (TN 10, Pos. 62)

*Think of it this way, our aim was to make our voices heard even in the furthest corners of the prison, so that we can communicate what is happening to the outside or to different places, it also removes some of the weight of the attack. And then you shout your slogans and experience torture. (TN 14, Pos. 40)*

*Long-term consequences* (in total 212 quotes): The interviewees reported different aspects of long-term psychological and psychosomatic consequences of their aversive experiences in Türkiye. This included a lack of trust as well as a sense of injustice. Several participants described stress symptoms, exhaustion, heaviness, listlessness, and rumination or inner tension. Some participants said they used cigarettes to calm themselves after arousal, some mentioned suicidal thoughts. Most interviewees stated that they experienced memory lapses and difficulties in concentrating, or other psychosomatic complaints, such as fatigue and headaches. Many reported loss of appetite and weight loss, insomnia, nightmares, and problems sleeping through the night. One participant described facial and body rashes due to psychosomatic

stress. Another participant mentioned frequent nausea and dizziness since the imprisonment and another interviewee severe back pain and a herniated disk.

> *I guess I must have been a little bit nervous . . . I've been distracted . . . I'm having troubles concentrating on anything . . . And the feeling of tiredness . . . (TN 4, Pos. 108)*

> I get angry all of a sudden. I have sudden crying fits. I mean, the slightest, the smallest thing can remind me of the past even if I'm laughing normally, at that moment I can immediately break away from that environment and go there. I can't help it; I can't stop it.

> Sometimes my brain stops, I can't think of anything. I'm in such a void, I force myself, I force myself, I force myself, I force myself, I force myself, I can't think of anything. That moment ends there. (TN 10, Pos. 166–167)

> *I cannot imagine any positive dreams. For example, let's say I go from here to the centre, I cannot dream of happiness, I say to myself 'the opposite will happen'. I feel anxious, as if the good things I want to do will always be the opposite. (TN 16, Pos. 166–168)*

*Hypervigilance and lack of trust* (46 quotes): Several interviewees reported that most often, they checked their surroundings for any possible threat and that they were alarmed when they spotted police officers, also outside of Türkiye. Some participants stated they avoided social contacts in Türkiye and no longer trusted anyone.

> *It's a very basic feeling of insecurity. For example, before I used to go to bed without closing my curtains, but after this incident I felt like someone was watching me all the time, and so you can't be comfortable at all . . . At first, I was very anxious, and I had to constantly check everything left and right. (TN 1, Pos. 94)*

> Sensitivity to sudden noises, if this glass falls down, for example. Anxiety about radio noise, sudden noises at night or if someone wakes me up suddenly. (. . .) I mean, I am in a state of panic, I am in a constant state of alertness. (TN 2, Pos. 73)

> *I didn't trust anyone, to be honest . . . Until I left Türkiye . . . (TN 21, Pos. 45)*

*Isolation, loneliness, and silence* (25 quotes): Many reported that in order to protect themselves or others from negative emotions and pain, they did not talk about their experiences state repression and torture. They said their relatives received only vague information about their detention and injuries.

> *I don't talk to anyone so that I don't make others feel bad, so that I don't make them feel bad. (TN 12, Pos. 91)*

> *So, I experienced the same stress at home, so I went into a room and closed the door so as not to hurt my children, so as not to upset them. This situation made me more and more isolated. (TN 16, Pos. 154)*

*Suppressing feelings and forgetting* (43 quotes): Some participants described how they avoided thinking about their aversive experiences, others mentioned they blocked out the news reports that could remind them of it. Some forgot parts of the torture experience or other abuses.

*Because of all this, I always want to escape from myself, I always want to run away from myself. Let me stay away from these problems, from these issues. (TN 16, Pos. 170)*

*Because people protect themselves and they do things. For example, as if it never happened, you freeze things that cause you severe trauma, as if it never happened, as if it hurts you so much that you freeze it and put it in the closet, as if it never happened. And after a while you start to believe it. (TN 20, Pos. 14)*

## Current situation

*Political Conviction and Religious Belief* (64 quotes): Most interviewees stated that their political or religious conviction helped them to overcome the aversive experiences they had and to move on. Many described that they were prepared by the stories of other political activists, which made it easier for them to process the events. It served as a resource to classify and process what had happened and to demand justice.

*Our experiences became more intensive with every second. That, of course, because of our political beliefs, and the work we did. But because we did it in full awareness of the consequences, we could be a little bit more prepared, (. . .) knowing that there might be problems. And at the same time, I can cope a little bit more myself because I see and I know that I am not the only one who has experienced it and that there are others who have experienced it more severely. I mean, there are others who have been through it worse, and if they have been able to get through it, you can get through it too, or we can get through it all together. (TN 2, Pos. 101)*

Actually, it is not us who are dishonoured . . . Those who let us experience this degrading treatment should also be the ones whose dignity is taken away. . . How can they look their children in the eyes? Actually, when you think about it, they are the ones who are dishonoured . . . (TN 14, Pos. 75)

*Psychologically I think these feelings help me. Let me put it this way, for example: if you ask me "do you regret it?", I don't regret it at all. Even if the same thing happened to me again, I think that in principle, ethically, as a human being, if I am doing something for humanity, I should go on doing it. (TN 21, Pos. 51)*

*Subjective well-being, work and social activities* (74 quotes): Many participants noticed performance impairments in themselves that they related to their experiences with torture and criticised their general state of health. Most participants reported they had no regular daily structure, especially those who lived in a refugee camp with an insecure asylum status. Several participants hoped to find work or their own apartment soon but said that this was more complicated due to the Covid-19 pandemic. In general, interviewees were socially integrated in Turkish communities and mentioned friends in Germany. Many reported taking care of themselves by going for walks, listening to music or reading books and watching videos, but some felt they didn't have the strength to do so at the moment.

*Sometimes I still feel like I'm watching a movie when I'm walking around in Germany, in the woods, at the lake, etc. . . Is it real or is it just another dream? Because I dreamt every day for a year. Every day I dreamt that I was leaving the country to start a new life with my family abroad. (..) So now I feel like I'm dreaming again. (TN 22, Pos. 48)*

I go for walks regularly, I try to eat, but no matter how much I do, this life in the refugee accommodation always stops me from doing something and I stay there. That's why I don't do anything. (TN 8, Pos. 178)

*What's happened has already happened. You don't see it as if nothing happened, but at least you can take a deep breath. But something is still squeezing my heart. You breathe, but your chest isn't full. (. . .) There's something missing. (TN 15, Pos. 24)*

*Plans for the Future* (34 quotes): About half of the interviewees felt optimistic about their future. Others expressed doubt about their general situation in Germany and whether they could achieve their goals. One participant said that he no longer made long-term plans, because all his plans had been interrupted when he had to flee Türkiye and he feared this to happen once again when the authorities cancelled his residence permit.

*Generally, I feel that my motivation to live has dropped a lot. Waking up in the morning, facing the day, experiencing the excitement of the day are things that I haven't felt for a long time, ever since I came here. . . (TN 18, Pos. 107)*

*What we're going to do here will always refer to there. That's what will always remain unresolved in my heart as long as I can't go back to Türkiye. Running down the street with the camera in my hand. Because that's what I want, because that's what I did, because that's what I learned . . . So, on the one hand we were the ones who experienced the torture, but on the other hand we were also the ones who somehow documented it. (TN 2, Pos. 141)*

## Discussion

Torture is a profound public health concern because it seeks to undermine not only the physical and emotional well-being of individuals, but also the coherence of entire communities. In this regard, the use of torture and other degrading treatment as state repression is a method to weaken a whole subpopulation. This research documents the emotional and physical reactions of survivors of torture and state repression that fled to Germany to seek asylum. Therefore, the presented study aimed at a) assessing personal experiences of torture and state terror in Türkiye between 2015–2018 and b) evaluating the mental health burden that arose due to these experiences.

In interviews based on the principles of the Istanbul Protocol, reports of twenty asylum-seekers from Türkiye convey a portrait of a suppressive state that uses illicit means to silence human rights activists and political opponents. Torture and inhuman treatment became a rising concern in detention centres after the failed coup attempt [23, 31, 33]. The methods described by various interviewees are consistent with evidence from the last decades about the use of unlawful treatment of dissidents, especially Kurds [4, 9, 12]. The depicted strategies to humiliate detainees were diverse and spanned from threats to unjustified dominant behaviour under police custody to electric shocks. In the descriptions of ill-treatment and sexualised violence by state officers and of the seemingly flawed system of routine medical controls, a power imbalance is clearly perceptible.

The fear and hypervigilance described by several participants appears to reach such levels that some suffer from PTSD, anxiety disorder, or panic disorder in the aftermath of the events. Some saw this as a contrast to the confident and self-righteous demeanour of members and supporters of the government. Several participants described symptoms of depression, like heaviness and rumination, and some describe suicidal thoughts. This is an alarming sign for

the mental health burden that is caused by state repression and torture. In light of our study, the need for more efforts to support refugees that have experienced extreme violence is evident.

The overall health of the interviewees seemed to be severely threatened after their experiences of torture and state repression. Some participants displayed signs of addiction from cigarettes as a form of self-medication to calm themselves, others suppressed their emotions or isolated themselves from family and friends. In the face of the experiences as well as the flight to Germany, immense exhaustion became apparent. The months of inner tension had left their mark on most interviewees. For many, political and religious beliefs were of great importance. They found comfort in the fact that their relatives and peers had already experienced imprisonment and police custody and the support they received by them. This way, they felt a sense of unity and purpose despite the violence that was inflicted on them. Their recovery was often limited by the current living situation under Covid-19, their insecure asylum status, and conditions in refugee camps. Most of them were socially integrated in Turkish communities in Germany.

## Limitations

First of all, the presented experiences of torture and state violence are derived from individual narratives and reflect the personal impressions and insights of the participants that have fled their home country due to life-threatening events. Of course, their reports are shaped by their individual perception of danger and past experiences with discrimination. Due to our qualitative approach, this study includes only a small number of participants and is far from representative. Nevertheless, we managed to interview Turkish citizens from various groups, such as Kurds, Alevis, and supporters of the Gülen movement.

When we conducted the first interviews, we realised that not all interviewees had been imprisoned, as formerly expected, but still every interviewee had experienced aversive and degrading treatment in Türkiye. All of them identified as survivors of torture, defined by them as illicit actions by the state against individuals or groups of people. According to the guidelines provided by the Istanbul Protocol, the severity of the pain and suffering is one of the key elements in defining torture, rather than the nature of the acts inflicted upon the person [35]. Therefore, the survivor perspective is considered to be crucial in the evaluation of aversive experiences [1].

## Conclusion

This study is an important contemporary testimony of a relevant period in Turkish history. The collected reports of torture and state repression and its psychological impact send an alarming sign for policy makers to address the prevention of human rights violations and the support needs of survivors from countries where torture and state repression is still practiced today.

## Supporting information

**S1 File. Quotes in the original language Turkish in order of appearance.**
(DOCX)

## Author Contributions

**Conceptualization:** Estella Alejandra Tambini Stollwerck, Ilkem Sarikaya, Christoph Nikendei.

**Data curation:** Estella Alejandra Tambini Stollwerck, Ilkem Sarikaya.

**Formal analysis:** Estella Alejandra Tambini Stollwerck.

**Investigation:** Estella Alejandra Tambini Stollwerck, Ilkem Sarikaya.

**Methodology:** Estella Alejandra Tambini Stollwerck.

**Project administration:** Estella Alejandra Tambini Stollwerck.

**Resources:** Ilkem Sarikaya, Kathrin Yen.

**Supervision:** Kathrin Yen, Christoph Nikendei.

**Writing – original draft:** Estella Alejandra Tambini Stollwerck.

**Writing – review & editing:** Estella Alejandra Tambini Stollwerck, Ilkem Sarikaya, Kathrin Yen, Hans-Christoph Friederich, Christoph Nikendei.

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
