## [Decision Letter · Decision Letter 0]

30 Jan 2024

PGPH-D-23-01899

The Psychological Impact of Torture and State Repression in Türkiye between 2015 and 2018:

Reports from Turkish Refugees Seeking Asylum in Germany

Dear Dr. Tambini Stollwerck,

Thank you for submitting your manuscript to PLOS Global Public Health. After careful consideration, we feel that it has merit but does not fully meet PLOS Global Public Health’s publication criteria as it currently stands. Therefore, we invite you to submit a revised version of the manuscript that addresses the points raised during the review process.

Please note that we have only been able to secure a single reviewer to assess your manuscript. We are issuing a decision on your manuscript at this point to prevent further delays in the evaluation of your manuscript. Please be aware that the editor who handles your revised manuscript might find it necessary to invite additional reviewers to assess this work once the revised manuscript is submitted. However, we will aim to proceed on the basis of this single review if possible. 

The reviewer has recommended that you cite specific previously published works. As always, we recommend that you please review and evaluate the requested works to determine whether they are relevant and should be cited. It is not a requirement to cite these works. 

We look forward to receiving your revised manuscript.

Kind regards,

Steve Zimmerman, PhD

PLOS Staff Editor

Journal Requirements:

Additional Editor Comments (if provided):

Reviewers' comments:

Reviewer's Responses to Questions

**Comments to the Author**

1. Does this manuscript meet PLOS Global Public Health’s publication criteria? Is the manuscript technically sound, and do the data support the conclusions? The manuscript must describe methodologically and ethically rigorous research with conclusions that are appropriately drawn based on the data presented.

Reviewer #1: Yes

2. Has the statistical analysis been performed appropriately and rigorously?

Reviewer #1: N/A

3. Have the authors made all data underlying the findings in their manuscript fully available (please refer to the Data Availability Statement at the start of the manuscript PDF file)?

Reviewer #1: Yes

4. Is the manuscript presented in an intelligible fashion and written in standard English?

Reviewer #1: No

5. Review Comments to the Author

Reviewer #1: The author(s) wrote a manuscript on The Psychological Impact of Torture and State Repression in Türkiye. I believe the study addresses a very interesting, timely and rarely studied topic which appeals to the readership of PLOS. I read it with great interest; however, the manuscript needs substantial revision and structural reframing to be considered for publication. There are a few issues need to be addressed before gets accepted for the publication.

1. The introduction needs to justify and articulate the focus and questions of this study from the relevant scholarship, policy, and practice perspectives using relevant literature as support. You need to develop research questions at the end of the introduction in alignment with the study focus. Also, there is limited citations from 2017-2023 that APA requires the most recent sources. I recommend to the author(s) to add a few the most recent sources in both literature review and discussion section.

Here are some most recent sources that might be very helpful;

https://www.amazon.com/Human-Rights-Turkey-Philosophy-Explorations/dp/303057475X

https://www.tandfonline.com/doi/full/10.1080/14782804.2023.2193876

https://www.ajqr.org/article/human-rights-conflicts-and-dislocation-the-case-of-turkey-in-a-global-spectrum-8266

https://www.tandfonline.com/doi/full/10.1080/14767724.2021.2017264

2. The theoretical/conceptual framework is addressed but missing justification and the author seems to use the Introduction/ literature review section as a framework. I think there needs to be a separate and strong section on the theoretical framework which forms a foundation for the study under the figure of framework. Without the theoretical framework, the justification of methodology and method of analysis remain vague and unclear.

3. Before moving on to Methods, it would helpful for the readers, if the authors to provide a brief discussion of their own context. This is based on a belief that who you are and how you involved with the topic and study will influence the presentation. We want readers to have a fair understanding of the researcher to make the fullest evaluation of the study and to have greater confidence in what they are about to read. Toward that end, please tell us your relationship to this inquiry. Who are you? What is your interest in this topic? What is your investment in this project? What are your intentions? In addition, your study does not appear to adhere to case study trustworthiness procedures.

4. The Discussion is much abbreviated. This section needs a description of your study aims and a brief summary of the findings as they relate to these aims, followed by a comprehensive summary of how your findings compare and contrast to those of previous researchers (including relevant citations) that I only see a few. This will give the paper greater consistency and structural coherence, leaving readers with a clear sense of the paper's aims and trajectory.

5. Please provide the limitations for the study

6. A few bullet points for implications of the study would be helpful for readers.

7. I am afraid there are some grammatical and English errors in this paper which make it quite hard to read in places and this is sufficient to make it un-publishable in its present form. The article itself is interesting but it is a difficult read given the issues above. I suggest authors always have their work edited by a professional English editor before submissions. Find someone who is a native speaker of English to proofread your paper

6. PLOS authors have the option to publish the peer review history of their article (what does this mean?). If published, this will include your full peer review and any attached files.

**Do you want your identity to be public for this peer review?** For information about this choice, including consent withdrawal, please see our Privacy Policy.

Reviewer #1: No

---

## [Decision Letter · Decision Letter 1]

30 Apr 2024

PGPH-D-23-01899R1

The Psychological Impact of Torture and State Repression in Türkiye between 2015 and 2018: Reports from Turkish Refugees Seeking Asylum in Germany

Dear Dr. Stollwerk,

Thank you for submitting your manuscript to PLOS Global Public Health. After careful consideration, we feel that it has merit but does not fully meet PLOS Global Public Health’s publication criteria as it currently stands. Therefore, we invite you to submit a revised version of the manuscript that addresses the points raised during the review process.

We look forward to receiving your revised manuscript.

Kind regards,

Jasmin Lilian Diab, Ph.D

Guest Editor

Journal Requirements:

2. Please send a completed 'Competing Interests' statement, including any COIs declared by your co-authors. If you have no competing interests to declare, please state "The authors have declared that no competing interests exist". Otherwise please declare all competing interests beginning with the statement "I have read the journal's policy and the authors of this manuscript have the following competing interests:"

Additional Editor Comments (if provided):

What I find missing from this piece is the framing of 'torture' and torture-related practices in Turkiye. How does the state define, control, or avoid these practices? While the arguments are strong, I think a nuanced understanding of torture remains missing, and framing this conversation against the backdrop of the wider political and human rights discourse/environment in Turkiye remains absent. An independent section on this to introduce the analysis is pivotal. Drawing more on the qualitative data collected would also enhance the rigor of the piece and this angle.

Reviewers' comments:

Reviewer's Responses to Questions

**Comments to the Author**

1. If the authors have adequately addressed your comments raised in a previous round of review and you feel that this manuscript is now acceptable for publication, you may indicate that here to bypass the “Comments to the Author” section, enter your conflict of interest statement in the “Confidential to Editor” section, and submit your "Accept" recommendation.

Reviewer #1: All comments have been addressed

2. Does this manuscript meet PLOS Global Public Health’s publication criteria? Is the manuscript technically sound, and do the data support the conclusions? The manuscript must describe methodologically and ethically rigorous research with conclusions that are appropriately drawn based on the data presented.

Reviewer #1: Yes

3. Has the statistical analysis been performed appropriately and rigorously?

Reviewer #1: Yes

4. Have the authors made all data underlying the findings in their manuscript fully available (please refer to the Data Availability Statement at the start of the manuscript PDF file)?

Reviewer #1: Yes

5. Is the manuscript presented in an intelligible fashion and written in standard English?

Reviewer #1: Yes

6. Review Comments to the Author

Reviewer #1: Authors are addressed my recommendation. Thanks

7. PLOS authors have the option to publish the peer review history of their article (what does this mean?). If published, this will include your full peer review and any attached files.

**Do you want your identity to be public for this peer review?** For information about this choice, including consent withdrawal, please see our Privacy Policy.

Reviewer #1: No

---

## [Editor Report · Decision Letter 2]

5 Jun 2024

The Psychological Impact of Torture and State Repression in Türkiye between 2015 and 2018: Reports from Turkish Refugees Seeking Asylum in Germany

PGPH-D-23-01899R2

Dear Estella Alejandra Tambini Stollwerck,

We are pleased to inform you that your manuscript 'The Psychological Impact of Torture and State Repression in Türkiye between 2015 and 2018: Reports from Turkish Refugees Seeking Asylum in Germany' has been provisionally accepted for publication in PLOS Global Public Health.

Best regards,

Jasmin Lilian Diab, Ph.D

Guest Editor